# Differentiating True and False Cinnamon: Exploring Multiple Approaches for Discrimination

**DOI:** 10.3390/mi14101819

**Published:** 2023-09-23

**Authors:** Giovana Feltes, Sandra C. Ballen, Juliana Steffens, Natalia Paroul, Clarice Steffens

**Affiliations:** Department of Food Engineering, Universidade Regional Integrada do Alto Uruguai e das Missões, Av. Sete de Setembro, 1621, Erechim 99709-910, Brazil; giovanafeltes@uricer.edu.br (G.F.); sandra-ballen@live.com (S.C.B.); julisteffens@uricer.edu.br (J.S.); nparoul@uricer.edu.br (N.P.)

**Keywords:** discrimination, physical–chemical analysis, instrumental analysis, authenticity, quality assurance, food industry

## Abstract

This study presents a comprehensive literature review that investigates the distinctions between true and false cinnamon. Given the intricate compositions of essential oils (EOs), various discrimination approaches were explored to ensure quality, safety, and authenticity, thereby establishing consumer confidence. Through the utilization of physical–chemical and instrumental analyses, the purity of EOs was evaluated via qualitative and quantitative assessments, enabling the identification of constituents or compounds within the oils. Consequently, a diverse array of techniques has been documented, encompassing organoleptic, physical, chemical, and instrumental methodologies, such as spectroscopic and chromatographic methods. Electronic noses (e-noses) exhibit significant potential for identifying cinnamon adulteration, presenting a rapid, non-destructive, and cost-effective approach. Leveraging their capability to detect and analyze volatile organic compound (VOC) profiles, e-noses can contribute to ensuring authenticity and quality in the food and fragrance industries. Continued research and development efforts in this domain will assuredly augment the capacities of this promising avenue, which is the utilization of Artificial Intelligence (AI) and Machine Learning (ML) algorithms in conjunction with spectroscopic data to combat cinnamon adulteration.

## 1. Introduction

Essential oils (EOs) are natural products obtained from different parts of plants (roots, bark, leaves, and flowers). They are hydrophobic liquids containing complex mixtures of aromatic compounds derived from the biosynthesis of secondary metabolites, serving as a defense against external agents. The components of EOs belong to different classes of compounds, such as monoterpenes and sesquiterpenes, alcohols, esters, aldehydes, ketones, and phenols, all of which possess various biological properties [1]. Currently, EOs have been widely used in the food, cosmetics, pharmaceutical, and chemical industries, as well as in perfumery [2].

One of the highly valued essential oils (EOs) by the food industry is cinnamon essential oil (CEO) from the *Cinnamomum* genus, which can be extracted from the root, bark, or leaves, each presenting distinct chemical compositions. Notably, cinnamon bark essential oil (EO) is rich in cinnamaldehyde, leaf EO contains eugenol, and root EO contains camphor [3]. CEO has been shown to possess antimicrobial [4,5], antifungal [6,7,8], anti-inflammatory [9,10], and antioxidant [11,12] activities. Consequently, CEO finds applications in culinary preparations due to its sweet and spicy aroma and flavor. It is commonly used in meat seasoning, baked goods, and pastries as an alternative preservative [13], and in chewing gum as a flavoring agent [14].

Cinnamon can be found on the market in two main species: *Cinnamomum verum* (true cinnamon or *Ceylon cinnamon*) and *Cinnamomum cassia* (syn. *Cinnamomum aromaticum*, false cinnamon). Due to its high market value, sweeter and milder flavor, and higher amounts of phenolic and aromatic compounds such as eugenol and cinnamaldehyde, true cinnamon is more challenging to obtain compared to false cinnamon. False cinnamon has a more astringent taste and contains a higher concentration of coumarin in its composition. Consuming large amounts of coumarin can lead to adverse health effects, thus justifying its lower cost in the market. As a result, true cinnamon is susceptible to fraud due to its quality and high value, and false cinnamon is often used as a substitute and/or adulterant, both in powdered form and as a CEO. 

In detail, in just over the last thirteen years, cinnamon has received significant attention by the scientific community. The search employed specific terms, including “true and false cinnamon”, “*Cinnamomum verum* and *Cinnamomum cassia*”, “authenticity of cinnamon essential oil”, or “cinnamon adulteration”, and focused on articles where these terms appeared in the field title, abstract, and keywords. The intention was to exclude publications unrelated to the subject matter. The search encompassed the entire period from the inception of the Scopus database up to 2010, with the final retrieval conducted in September 2023. It considered original articles and review articles as eligible documents for our analysis. Data were extracted from publications across diverse fields of knowledge and originating from various countries. We included original articles and reviews published in English. Conversely, other publication categories indexed on Scopus were excluded from our analysis, as publications in languages other than those mentioned were limited in the context of our research. Figure 1 illustrates the upward trajectory of scientific publications related to cinnamon over time. This is evidenced by a total of 120 scientific papers published in recent years.

The assurance of quality, safety, and authenticity of EOs is a paramount concern due to their intricate compositions. EOs, derived from various plant parts, encompass a diverse array of aromatic compounds, each contributing to their distinct aromas and potential therapeutic properties. As these oils find their way into numerous products in industries ranging from food and cosmetics to pharmaceuticals, ensuring their purity and legitimacy becomes imperative to safeguard consumer well-being. 

A plethora of techniques have been detailed in the scientific literature to address the multifaceted nature of EOs. These methodologies span a spectrum from traditional organoleptic assessments, which involve human sensory evaluations, to more advanced and precise analytical methods. Physical, chemical, and instrumental approaches play pivotal roles in verifying the genuineness of CEOs. 

Spectroscopic techniques, such as infrared and nuclear magnetic resonance spectroscopy, provide insights into the molecular structure of EOs, aiding in their identification and authentication. Chromatographic methods, such as gas chromatography-mass spectrometry (GC-MS) and high-performance liquid chromatography (HPLC), enable the separation and quantification of individual compounds within the complex EO matrices. These techniques serve as valuable tools for pinpointing specific markers that can distinguish authentic CEOs from adulterated or counterfeit versions. 

Nevertheless, the successful application of these techniques demands skilled personnel, extensive equipment, and a considerable time investment. Moreover, some approaches may inadvertently alter the CEO sample or are resource-intensive, posing challenges to their widespread use.

In response to these limitations, there is a growing need for the development of novel analytical methods that align with the requirements of cost-effectiveness, simplicity, and reliability. Such methods could potentially bridge the gap between the complex analysis needed for EO authenticity and the practical feasibility demanded by the industry.

## 2. *Cinnamomum* sp.

Cinnamon is a spice belonging to the genus *Cinnamomum* in the Lauraceae family, and it is widely used across various cultures around the world [15]. The name “cinnamon” is derived from the Greek word, meaning “sweet wood” [16]. It is a perennial tree that can grow to heights of 7 to 10 m, although it can also be cultivated as a shrub, reaching less than 3 m in height. Cinnamon thrives in tropical, warm, and humid climates, and it becomes ready for harvest after about three years of growth. It features dark green leaves, small white-yellowish flowers, and purple fruits that contain a single seed [3]. 

Around 250 to 350 species of cinnamon have been identified and distributed across North America, Central America, South America, Southeast Asia, and Australia. Among these species, four are considered of greater importance and are commonly used for obtaining the spice: *Cinnamomum zeylanicum* Blume (also known as *C. verum*), *Cinnamomum aromaticum* (or *C. cassia*), native to China, *Cinnamomum burmannii*, native to Indonesia, and *Cinnamomum loureiroi*, native to Vietnam [17,18].

Its consumption is associated with health benefits such as antimicrobial activity [19], antioxidant properties [20], anticancer effects [21], and glucose control in diabetes [22]. Cinnamon is utilized in foods, seasonings, cosmetics, and medications, and is available in various forms, such as whole material, ground, extracts, or essential oils obtained from the leaves and bark. However, the consumption of cinnamon may also lead to adverse health effects. *Trans*-cinnamaldehyde, also known as cinnamaldehyde, the main component of cinnamon bark, can result in skin sensitization and cause contact dermatitis [23]. Cinnamic acid is also known to induce hypersensitivity upon contact [24]. 

True cinnamon native to Sri Lanka, also known as *Ceylon cinnamon*, includes the species *C. verum* and *C. zeylanicum*, and false cinnamon, which have diverse origins such as China, South America, and Indonesia, and include the species *C. cassia*, *C. aromaticum*, *C. burmannii*, and *C. loureiroi* [25] (Table 1). 

The aroma and taste of true cinnamon are soft and sweet, and its color is light brown, whereas fake cinnamon is darker, in addition to having a stronger, astringent, and spicy flavor [26]. In addition, they have distinct characteristics regarding the composition of phenolic compounds, with true cinnamon being rich in aromatic and phenolic compounds, such as cinnamaldehyde and eugenol, while false cinnamon has higher amounts of coumarin (2000 to 5000 mg/kg), approximately a thousand times higher than those found in true cinnamon (2 to 5 mg/kg), and tannins in the bark, which explains the astringent taste [25,27,28,29].

**Table 1 micromachines-14-01819-t001:** Differences between *Cinnamomum verum,* and *Cinnamomum cassia* species.

	*C. verum*	*C. cassia*	*C. burmannii*	*C. loureiroi*
Country where it originates	Sri Lanka	China	Indonesia	Vietnam
Flavor	Mild Sweet	Bitter Spicy	Spicy	Sweet spicy
Color	Light reddish brown	Dark reddish brown	Dark reddish brown	Dark reddish brown
Coumarin content (g/kg)	0.017	0.31	2.15	6.97

Font: Adapted from Kawatra and; Rajagopalan [30].

Coumarin is an anticoagulant agent that can pose serious health risks due to its hepatotoxic and carcinogenic effects in animals. For this reason, health agencies have established restrictions regarding the tolerable daily intake of coumarin, thus determining the daily intake at 0.1 mg/kg/day, and consequently the consumption of *C. cassia*, guaranteeing its safe use [31,32].

The European Regulation (EC) No 1334/2008 [33] established specific coumarin maximum limits as follows: 50 mg/kg for traditional and/or seasonal bakery products with cinnamon mentioned in their labeling, 20 mg/kg for breakfast cereals, including muesli, 15 mg/kg for fine bakery products (excluding traditional and/or seasonal bakery products with cinnamon in the labeling), and 5 mg/kg for desserts. 

In general, *C. loureiroi* contains high levels of coumarin, which can cause adverse side effects for the consumer, including liver damage. True cinnamon, on the other hand, is highly vulnerable to fraud due to its added value and higher quality compared to false cinnamon [34]. 

*C. verum* is a highly prized spice and is considered superior to false cinnamon, which is commonly found and cheaper. Visually, the bark cinnamon appears in the form of rolled cylinders. However, the true cinnamon has a light reddish-brown color rolled in several layers, while the others are in dark reddish-brown tones, hard, and rolled in only one layer [30]. 

The different parts of cinnamon bark, leaf, or powder, have differences in composition (Figure 2), which can be used for quality control. The bark, for example, contains natural antioxidants, while the branches are used to treat inflammatory diseases. Cinnamon bark powder, when added with other medications, can slow down the deterioration process of some heart conditions [35,36,37].

### 2.1. Cinnamon Essential Oil

CEOs are complex mixtures of aromatic products from the secondary metabolism of plants, normally produced by secretory cells or groups of cells from different parts of the plant, such as stems, roots, leaves, flowers, and fruits [38,39]. The essential oil content may vary according to the species, physical form of the sample, part of the plant used (Table 2), geographical origin, and stage of development of the plant [3]. 

The constituents of CEOs can belong to several classes of compounds, with emphasis on terpenes and phenylpropenes, which are the classes of compounds commonly found. Monoterpenes and sesquiterpenes are the most frequently found terpenes in EOs, as well as diterpenes, and minor constituents [40].

Different parts of cinnamon, bark, leaves, branches, fruits, and roots can be used for the production of essential oils by distillation and oleoresins by solvent extraction [41]. The volatile components of the EO are present in all parts of the plant (Table 2) and can be classified into monoterpenes, sesquiterpenes, and phenylpropenes, with main constituents such as *trans*-cinnamaldehyde (bark), eugenol (leaves), and camphor (root) [3,42].

**Table 2 micromachines-14-01819-t002:** Volatile compounds present in the bark and leaf of *C. verum* and *C. cassia* essential oils.

Compounds	Content (%)			
	Bark		Leaf	
	*C. cassia*	*C. verum*	*C. verum*	*C. cassia*
1,3-dimethyl-benzene	0.23	0.15	-	-
Styrene	0.19	0.14	-	-
Benzaldehyde	0.41	0.29	0.05	0.10
Camphene	0.35	0.21	-	-
Acetophenone	0.96	tr	-	-
β-Pinene	0.15	0.44	-	-
Linalool	0.68	-	-	-
Camphor	0.97	0.53	-	-
Benzene propanal	0.64	0.53	-	-
Borneol	0.19	0.12	-	-
*Cis*-cinnamaldehyde	1.95	2.29	-	-
*Trans*-cinnamaldehyde	77.21	74.49	16.25	30.65
Eugenol	0.21	7.29	79.75	-
Geranyl acetate	0.14	0.12	-	-
Benzene,1-(1,5-dimethyl-4-hexenyl)-4-methyl	0.37	0.15	-	-
Cinnamyl acetate	0.14	0.49	-	-
α-muuroleno	0.47	0.11	-	-
3-Methoxy-1,2-propanediol	-	-	-	29.30
Cinnamyl alcohol	-	-	0.07	0.65
Acetaldehyde	-	-	-	0.47
o-Methoxy cinnamaldehyde	-	-	-	25.39
Coumarin	-	-	0.05	6.36

tr—trace; Font: Chairunnisa; Tamhid; Nugraha, [43]; Li; Kong; Wu [44].

Cinnamaldehyde is a yellowish oily liquid responsible for the strong odor and sweet taste of cinnamon [17] and the main constituent of EO. It is recognized as safe by the United States Food and Drug Administration and the Association of flavor and extract manufacturers, receiving status A, that is, it can be used in food [45]. Its commercial use is limited due to its low solubility in water and sensitivity when exposed to light and air for prolonged periods [46]. Friedman; Kozuke; Harden [47] evaluated the stability of *trans*-cinnamaldehyde present in EO at different temperatures and observed that around 60 °C the compound was decomposed. Das, Gitishree et al. [48] report that cinnamaldehyde has a biological effect and is quickly oxidized into cinnamic acid, and as a product of its degradation, benzoic acid is excreted by the urinary system. 

Due to cinnamic acid and cinnamaldehyde, cinnamon has protective effects against cardiotoxicity produced by the compound isoproterenol [49]. In addition, cinnamon is associated with the inhibition of fatty acids such as arachidonic acid, which has an inflammatory effect. The compound eugenol, identified in cinnamon extracts, has an antioxidant effect, helping to inhibit lipid peroxidation and the generation of reactive oxygen species [50]. 

Eugenol is the main volatile compound of OECF [51]. It is an aromatic substance with a pleasant odor and taste, belonging to the class of phenylpropanoids [52]. It is usually found as a yellowish oily liquid [53]. Like cinnamaldehyde, eugenol is also recognized as a safe food by the Food and Drug Administration [54]. Eugenol has antimicrobial [4,5], antifungal [6,7], anti-inflammatory [9,10], and antioxidant [11,12] activities. Despite having good biological properties, eugenol has low solubility in water [55], and sensitivity to light [54].

CEO contains some vital bioactive components in the form of terpenes and aromatic compounds, giving it remarkable biological properties. Thus, CEO has been widely used as a raw material in the medicine industry, as natural additives, condiments, and flavorings in the food industry, and in perfumery [42,56].

The main responsibility for the biological activities is often attributed to the major compounds present in the oil, such as the volatile fraction and the phenolic compounds [57]. In addition, biological activities can also be related to the joint contribution of different compounds, with minor components producing a synergistic effect with the others [3,58]. Table 3 provides an overview of the diverse biological activities associated with various species of Cinnamomum. The compounds found in cinnamon have demonstrated a wide range of biological effects, including antimicrobial, anti-inflammatory, antioxidant, insecticidal, and antidiabetic activities. This compilation highlights the multifaceted potential of Cinnamomum in various fields, from traditional medicine to modern pharmaceutical research, pest control, and diabetes management.

CEO can be used in culinary preparations due to its sweet and spicy aroma and flavor. It is generally used in seasonings for meat, fish, sauces, roasts, and beverages; in bakery and pastry products as an alternative to preservatives [13]; and in chewing gum as a flavoring agent [14]. With the growth in demand in the food, perfumery, and cosmetics industries, the market for natural aromatic raw materials is expanding exponentially to meet your needs. To meet this search for aromatic products, research has focused on compounds from biotechnological processes used in the production of aromas and fragrances from other plant origins [80].

### 2.2. Adulteration of Cinnamon Essential Oils (CEOs)

With the growth in demand in the food, perfumery, and cosmetics industries, the market for natural aromatic raw materials is expanding exponentially to meet your needs. To meet this search for aromatic products, research has focused on compounds from biotechnological processes used in the production of aromas and fragrances from other plant origins [80]. 

Intentionally altered products with hidden properties or quality and incomplete and unreliable information define adulteration. In general, counterfeiting involves actions that deteriorate the specific properties of the products while maintaining their characteristic indicators, such as appearance, color, consistency, and aroma [81]. 

Currently, the authenticity of food and food ingredients is a major challenge, as it is often related to fraud. Food adulterations have been occurring for a long time, with the difference that they have been improved over the years, accompanying or even advancing in the field of research into new methods. A greater number of adulterations among raw materials are found in spices, edible oils, honey, milk and its derivatives, fruits and fruit juice, coffee, flour, and meat products [26].

Among the raw materials, cinnamon is one of the spices commonly adulterated. These adulterations can include species (mixing different species of Cinnamomum), origin (incorrectly labeling the country of origin), additives and fillers (adding other substances such as starch, sawdust, or other spices), and oil (adulterating EOs derived from cinnamon with synthetic or cheaper oils).

Due to its high added value and high cost, true cinnamon is highly prone to adulteration with false cinnamon species, which are of lower quality and cheaper, causing potential health risks and making the product unsafe for the consumer. This practice is usually carried out in powder form, which makes it difficult to discriminate between cinnamons as their characteristics are lost during the process [29].

Furthermore, true cinnamon can be adulterated with other types of spices, such as clove and chili powder, and clove and cinnamon oil, as reported by Gopu et al. [82]. Cinnamon fraud triggered research for the development of new, more accurate, reliable, and sensitive analytical methods in order to identify and quantify potential adulterants in true cinnamon more quickly and efficiently, ensuring food safety. Among the methods developed, chromatography is based on the determination of the main active compounds of cinnamon or adulterants, called marker compounds, such as cinnamaldehyde, eugenol, linalool, and coumarin, among others [82]. 

Cinnamon adulterations do not only occur in the form of powder, frauds are also found in CEO by mixing other compounds or even false cinnamon species. Through physical–chemical and instrumental analyses, the CEOs purity can be proven through qualitative and quantitative analyses, that is, determining the constituents or identifying the compounds present in the oil, respectively [81]. 

To prevent the entrance of adulterated cinnamon products into local markets, the regulatory sector takes several steps: product testing and certification are performed by implementing regular product testing to verify the authenticity and quality of cinnamon products; certifying authentic products with recognized standards can help consumers identify genuine products; labeling regulations require accurate information about the species, country of origin, and any additives or fillers in the product; clear and transparent labeling helps consumers make informed choices; traceability systems that track the supply chain of cinnamon products from production to market, which can help identify and eliminate adulterated products at different stages; import controls and inspections to ensure that products entering the country meet regulatory standards, which includes checking for proper documentation and compliance with labeling regulations; public awareness and education about the different types of cinnamon and their characteristics—informed consumers are less likely to purchase adulterated products; penalties and enforcement for those found guilty of adulteration, and rigorous enforcement of these penalties, which can act as a deterrent to unethical practices; collaboration with industry and associations to establish self-regulation practices and codes of conduct that promote authenticity and quality; and international cooperation with international regulatory bodies and other countries to share information and best practices in combating cinnamon adulteration, especially when products are imported.

For the regulation of EOs, the molecules must come from the raw material of the reference plant to be considered and labeled as 100% pure and natural [80]. Adulteration of CEOs can be divided into four types [81]: essential oil diluted with a solvent that has similar physicochemical characteristics, such as vegetable oils or organic solvents; cheaper CEO, but similar in origin or chemical composition, mixed with authentic CEO; unique natural or synthetic compounds added to mimic aromatic characteristics or composition; and/or substituting with low-value or blending CEOs.

As they are complex matrices, the EOs need to be analyzed by different techniques to ensure quality, safety, and authenticity, in addition to ensuring safety for consumers. As a result, a wide range of techniques have been reported, including organoleptic, physical, and chemical methods. However, these techniques, for the most part, require specialized people, have a high investment cost, are time-consuming, and some degrade the samples [81,83].

Molecules produced from natural reagents, also known as semi-synthetic compounds, can be used to adulterate specific EOs [80]. Cinnamaldehyde molecules can be produced from the benzaldehyde found in bitter almonds and used as an adulterant [84]. There are several scams associated with bitter almond and CEO, and since the false origin is cheaper than the Ceylon origin, blends between these CEOs are common. Despite this, it is possible to detect this type of adulteration by analyzing differences in composition between EOs or by spectroscopic analysis [26,28,80]. 

Adulterations along the food chain present a health hazard, and continuous vigilance is fundamental in terms of food safety, regarding research and development of analytical methods to detect adulterations and contamination in food from the raw materials used [26].

## 3. Methods to Detect Compounds and Adulterations

Figure 3 provides a summary of the methods applied to detect compounds and cinnamon adulterations based on cost, quality, and diagnostic efficacy. Numerical values have been assigned to each method (low = 1, moderate = 2, and high = 3) in relation to the cost, quality assessment, and diagnostic efficacy. The effectiveness of these methods can vary depending on factors such as the expertise of the personnel conducting the analysis, the quality of reference samples, and the specific type of adulteration. Additionally, other factors such as speed, scalability, and regulatory compliance may also play a role in choosing the appropriate method for a given scenario.

### 3.1. Physical

The physical methods involved in the detection of adulterants are based on macroscopic and microscopic techniques and the physical property examination of spices. Macroscopic and microscopic techniques are used to recognize the structure of the surface, color, shape, and presence of foreign materials. Together with sensory analysis, it is possible to identify aroma and flavor. Several characteristics of plants, such as size, shape, texture, morphology, and tissues, are used as a basis for detecting adulterants for different parts of the plant, such as roots, leaves, stems, flowers, and fruits. Generally, these characteristics resemble real samples and are evaluated individually; the results found must be authenticated through microscopic methods [85,86].

The microscopic analysis aims to observe cellular, structural, and internal tissue characteristics of the food matrix, being more effective in powdered samples where the macroscopic characteristics have little effectiveness [85].

Kumar [87] examined leaves of *Cinnamomum malabatrum* sold as *Cinnamomum tamala* by macro and microscopic analysis in order to authenticate the samples. The powder was evaluated microscopically, while some dried leaves were preserved whole for histological study. As a result of the macroscopic analysis, there were differences between the leaves, mainly in the variation in the size of the leaf blade and petiole, on the surface, and on the veins. The microscopic analysis of the powder sample provided information on the different tissues that make up the petiole, proving to be important in differentiating the two species. The macro and microscopic analyses of the leaves obtained from the market were in agreement with the authentic sample of *C. malabatrum*.

Jeremić et al. [88] used macroscopic and microscopic methods in the analysis of cinnamon powder and bark from the Serbian market. Five cinnamon samples (two powder and three bark) were purchased from local grocers and stores; the bark samples were observed under a stereomicroscope, and the powder was examined under a binocular microscope. The macroscopic identification of the bark showed differences between the evaluated *C. burmannii*, *C. verum*, and *C. cassia* species. *C. cassia* has a hard and thick layer, while *C. verum* is composed of several layers of soft and thick bark. In the microscopic evaluation of cinnamon powder, the *C. burmannii* sample was easier to distinguish due to the presence of calcium oxalate. *C. verum* had a greater number of sclereids present alone or in small groups, in addition to many grains of starch and fiber. *C. cassia* showed a large amounts of starch granules and larger fiber diameter. The average diameter found was 69.60 and 52 µm for *C. cassia* and *C. verum*, respectively. Thus, the diameter of the fibers can be used as a parameter for verifying authenticity.

Binitha et al. [89] detected adulterations in 12 commercial samples from Kerala (A, B, C, D, E, F, G, H, I, J, K, L, and M) of *C. verum* using macroscopic, microscopic, and high performance thin layer chromatography (HPTLC) methods for coumarin quantification and analysis of heavy metals. In the samples analyzed macroscopically, five were similar to *C. verum* and six were similar to false cinnamon, which could be *C. cassia* or *C. burmannii*, and a single sample was different from all and was considered to be the species of *C. malabatrum*. All samples were identified as cinnamon species. The samples were within the permitted limit for heavy metals. The coumarin content was observed in greater quantity in samples A, B, and C, at 4.56, 11, and 9.85%, respectively, and samples D, E, G, and J were considered from *C. verum* with low coumarin content. Thus, the analyses were of great importance to detect the adulteration and authenticity of the cinnamon samples.

### 3.2. Sensorial

Sensory evaluation is used to measure, analyze, and interpret reactions perceived by the senses of sight, smell, taste, touch, and hearing. It is extremally important to discriminate the attributes of complex beverages and foods (such as wine, coffee, chocolate, dairy-free beverages, and EOs, among others). Sensorial techniques that use the olfactive human system can be used to evaluate or distinguish samples, but a trained panel is required, and the process is time-consuming and costly.

The aromatic plants have been known to give food its most intuitive sensory qualities, such as color, aroma, and taste [90]. Aromatic plants can be detected by olfactory perception. In humans, for example, the sense of smell is induced when the volatile substance stimulates the olfactory nerve in the nasal cavity. The olfactory system is the connection between the nasal cavity and the olfactory bulb, where odor molecules in the nasal mucosa come into contact with receptor cells, which, in turn, undergo an olfactory cascade reaction to the central nervous system and respond [91].

CEOs are composed of various volatile chemical components, such as terpenes, esters, alcohols, etc. The constituent differences lead to the aroma type. Humans have different perceptions of odors, which may influence the potential effects of those functional odors, mainly due to individual differences [92].

In the literature, there are studies that report that cinnamon, for example, is used as a flavoring agent in foods and beverages [93] and also in medicine and pharmacology [94]. Ochanda et al. [95] verified that the addition of cinnamon powder to purple tea increased consumer acceptability. However, Dima and Dima [96] related that the use of plant extracts in food enrichment is limited due to the sensitivity of the olfactory system and taste sensors of the consumer.

Jiménez-Redondo et al. [97] evaluated the effect of the addition of *C. cassia* and *C. verum* powders to sugar-free yoghurt at 0.5% and 1.5% concentrations. The addition of 0.5% cinnamon before yogurt fermentation allowed for high scores on consumer acceptability. Also, Suliman et al. [98] studied the addition of *C. cassia* powder in yogurt, and the results indicated that cinnamon is effective against pathogenic bacteria, and the sensory attributes were rated slightly lower, except taste.

Sudarsh and Müller-Maatsch [99] evaluated ground samples from different origins (Indonesia, Madagascar, Sri Lanka, Tanzania, and Vietnam) of the different varieties (*C. cassia*, *C. burmannii*, *C. zeylanicum*, and *C. loureiroi*) in relation to their sensory profiles in the aromatic compound coumarin. The volatile profiles of samples collected from *C. cassia*, *C. burmannii*, and *C. loureiroi* contained more cinnamaldehyde (81–95%), and eugenol (2–9%) from *C. zeylanicum*. The variations in volatile profiles between samples align with the potential for distinguishing between these varieties through sensory analysis or the analysis of volatile organic compounds.

### 3.3. Analytical

Analytical methods are based on the chemical composition or organic components present in the sample being used for identification and authentication. Following this, an array of analytical methods come to the forefront, standing as stalwarts in the realm of cinnamon analysis. These methods, meticulously designed and refined, navigate the intricate pathways of chemical intricacies to unravel the true essence and authenticity of cinnamon samples. Through these techniques, researchers and analysts delve deep into the molecular landscape, extracting insights that differentiate the genuine from the adulterated, ensuring the integrity of this prized spice.

#### 3.3.1. Spectroscopic

Spectroscopic techniques are used to analyze the composition and structure of spices, with emphasis on ultraviolet and visible, infrared, vibrational, fluorescence, Raman, mass spectroscopy, nuclear, and infrared magnetic resonance techniques. These techniques are based on the interaction of electromagnetic radiation with the sample, so they are fast, non-destructive, non-invasive, and highly sensitive [85].

Near-infrared spectroscopy (NIRS) is a non-destructive analytical technique that uses the absorption of near-infrared light by organic molecules to provide information about the composition of a sample. Infrared spectroscopy includes the visible to near-infrared (Vis/NIR) and mid-infrared (MIR) regions of the electromagnetic spectrum. NIR and MIR spectroscopy cover a range from 800 to 2500 nm and 2500 to 25,000 nm, respectively [86]. Thus, spectroscopy has been used as a method to confirm authenticity and possible adulteration.

In this context, NIRS can be used to verify and evaluate its quality by analyzing its chemical composition. Cantarelli et al. [26] evaluated 120 cinnamon samples, including 30 of *C. verum*, 30 of *C. cassia*, 30 adulterated samples 75/25 (true/false), and 30 adulterated samples 90/10 (true/false). The combination of NIRS-PLS-DA (NIR spectroscopy—partial least squares discriminant analysis) and NIRS-PNN (NIR spectroscopy—probabilistic neural network) showed excellent results in the detection and discrimination of samples, with a mean discrimination of 99.25% and 100%, respectively.

Cruz-Tirado et al. [100] used near-infrared hyperspectral imaging (NIR-HIS) combined with soft independent modeling class analog (SIMCA), partial least square discriminant analysis (PLS-DA), and support vector machine (SVM) classification chemometrics tools to differentiate cinnamon sticks from two species, *C. verum* and *C. cassia* (true and false, respectively), without grinding the samples. Cinnamon sticks from diverse geographical origins, including India, Peru, Brazil, and Sri Lanka, were gathered for the study. Additionally, samples lacking specific species identification were also included in the analysis. The principal disparities observed within the spectra are intricately linked to the fluctuating presence or absence, as well as the concentration variability, of phenolic compounds in both cinnamons. Notably, coumarin emerges as a significant marker for *C. cassia*, while catechin assumes prominence for *C. verum*. These distinctive spectral fingerprints shed light on the underlying chemical composition, unraveling the unique identities of these cinnamon varieties. SIMCA analysis did not provide good separation between cinnamon species, while PLS-DA and SVM showed similar performance, correctly classifying more than 90% of the samples according to species.

In a study by Castro et al. [101], a powerful combination of NIR spectroscopy alongside data-driven soft independent modeling of class analogy (DD-SIMCA), multivariate curve resolution-alternating least squares (MCR-ALS), and PLS techniques was harnessed to discern and quantify adulterants such as false cinnamon, cloves, and black pepper within authentic cinnamon samples. Impressively, all applied chemometric methods yielded gratifying outcomes. DD-SIMCA exhibited a classification success rate of 100%, marked by sensitivity and specificity values also reaching a perfect 100%. MCR-ALS showcased its capacity to reconstruct spectral profiles across various samples, highlighting its efficacy in distinguishing differences. Moreover, every PLS model demonstrated a p value surpassing 0.95, emphasizing the robustness of these models in terms of predictive accuracy. This integration of advanced spectroscopic and chemometric methodologies underscores their potential for enhancing the accuracy and reliability of cinnamon authenticity assessments.

Sudarsh and Müller-Maatsch [99] employed three different techniques, UV-Vis, NIR, and fluorescence (FLUO), to authenticate cinnamon samples in terms of differentiating between varieties, origin, and coumarin content. Remarkably, all three techniques demonstrated their efficacy in detecting the authenticity of the samples. By UV-Vis and FLUO, the samples can be categorized according to variations, origins, and coumarin content. The coumarin content is further divided into three groups: Group 1 comprises samples with less than 70 ppm of coumarin from *C. zeylanicum*; Group 2 includes samples with coumarin content ranging from 70 to 3000 ppm; and Group 3 encompasses samples with over 3000 ppm of coumarin. Notably, Groups 2 and 3 encompass samples from *C. loureirii*, *C. burmannii*, and *C. cassia*. Of the three methods used, UV-Vis and NIR were superior to FLUO, and the UV-Vis method was more effective in discriminating coumarin.

Shawky [32] detected adulterants in powdered *C. verum* using NIR spectroscopy combined with soft independent modeling of class analogy (SIMCA) and partial least squares regression (PLSR). The study involved a comparison between the NIR spectra of the samples and the individual spectra of key compounds, including cinnamaldehyde, eugenol, and cinnamyl acetate. Noteworthy similarities were identified between the spectral characteristics of these compounds and those found in the samples across multiple bands. Variations were detected in specific spectral regions, approximately between 6100 and 6020 cm^−1^ (associated with the stretching of -COC-), and 5970 and 5900 cm^−1^ (linked to the vinyl group and -CH aromatic). These differences were attributed primarily to the presence of coumarin and eugenol compounds in the samples. The fusion of NIR spectroscopy with multivariate analysis proved to be a viable approach for ensuring the quality control of powdered cinnamon, being observed a clustering pattern with a correlation with the aromatic compounds (cinnamaldehyde and eugenol), which are abundant in *C. verum* and found in smaller quantities in *C. cassia*. Additionally, the presence of coumarin, unique to *C. cassia*, appeared to play a significant role in this correlation. These findings were based on the identification of their characteristic functional groups, which were noted as the most prominent loadings in the analysis.

Lopes et al. [25] identified chemical markers for differentiating true and false cinnamon using HPLC and MIR spectroscopy methods from commercial samples from three countries (Brazil, Sri Lanka, and Paraguay). By HPLC, it was possible to observe that the true one had a high concentration of eugenol (6.53 mg/g), cinnamaldehyde (71.64 mg/g^−1^) and antioxidant capacity by DPPH (0.79% inhibition), total phenolic constituents (73.26 mg GAE/g), and total flavonoid constituents (45.07 mg CE/g), low concentration of coumarin (2.48 mg/g) and quercetin (13.23 mg/g), and similar values of epicatechins (13 mg/g), caffeic acids (16.09 mg/g) and catechin (10.93 mg/g). The spectra obtained with the MIR showed differences in minor compounds, such as phenolic compounds, which are found in higher amounts in true cinnamon. The 3500–3200 cm^−1^ band is associated with phenolic compounds (epicatechin, catechin, quercetin, caffeic acid, and eugenol), followed by a band of medium intensity of 1475–1450 cm^−1^, being associated with cinnamaldehyde and eugenol. The PCA analysis showed the separation of the two groups of samples studied, and the PLS-DA was effective in differentiating the samples, being 94.44% for a composition (HPLC) and 100% for MIR spectroscopy. The two methods used were effective in differentiating true and false cinnamon.

Lixourgioti et al. [102] evaluated the detection of adulteration of two species of cinnamon in two scenarios: A with *C. cassia* adulterated, and B with *C. verum* adulterated, both at levels of 1 g/100g to 99 g/100g. The samples evaluated were from Indonesia, China, India, Sri Lanka, Vietnam, and the Seychelles. Using gas chromatography ion mobility spectroscopy (GC-IMS), the *C. verum* demonstrated a greater number of VOCs (linalool, n-nonanal, benzaldehyde, heptanal, hexanal, 1-butanol, and pentanal). By FTIR analysis, in general, it is possible to observe the spectral fingerprints of *C. cassia* and *C. verum* are quite similar, each containing approximately 15 to 16 peaks. Notably, the cinnamon sample exhibited a reduced number of peaks compared to both *C. verum* and *C. cassia*. Discriminating the spectra using PCA, it was observed that when dealing with admixtures that have low levels of adulteration (below 10%), there is a noticeable overlap with the parent class. This overlap underscores the challenge of the analytical task and highlights the limitations of spectroscopic techniques in effectively distinguishing such low mixing ratios.

Yasmin et al. [29] determined by FT-NIR and FT-IR the spectral differences of true and false cinnamon samples and mixtures of the species as an adulterant combined with partial least squares regression (PLSR). The PLSR model combined with the FT-NIR spectroscopic technique and pre-processing methods showed greater accuracy with R^2^ of 0.97. The method developed after pre-processing can be used to detect false cinnamon powder in true cinnamon powder.

Farag et al. [27] used nuclear magnetic resonance (NMR) spectroscopy to distinguish two cinnamon species, *C. verum,* and *C. cassia*, and microscopically examined the bark. By microscopic analysis, the *C. verum* did not present a layer of cork, differentiating it from the *C. cassia*. Combining other methods, the PCA slightly differentiated, and when applied, OPLS-DA metabolic patterns correlated to the studied species were identified, clearly showing the discrimination between *C. verum* and *C. cassia*. From NMR spectroscopy, eugenol was identified in samples of *C. verum*. Thus, the approach used was feasible to analyze the cinnamon samples.

Avula et al. [103] used a direct analysis in real-time quadrupole time-of-flight mass spectrometry (DART-QToF/MS) technique for the qualitative identification and confirmation of chemical components in various samples of four different species of cinnamon (*C. loureirii*, *C. aromaticum*, *C. burmannii,* and *C. verum*). The results of the analysis revealed distinct chemical compositions in different types of cinnamon: *C. loureirii* showed a notably high percentage of *trans*-cinnamaldehyde, *C. aromaticum* contained abundant quantities of methyl cinnamate and guaiene (C_15_H_24_), which were more prevalent compared to *C. loureirii*, *C. burmannii*, and *C. verum*, and *C. burmannii* exhibited significant amounts of coumarin in comparison to the other three types of cinnamon. By analyzing DART-MS data, specific marker compounds were discovered that could serve as discriminators for distinguishing between ‘true cinnamon’ and other cinnamon samples.

These studies collectively contribute to the authentication and differentiation of ‘true cinnamon’ (*C. verum*) from other cinnamon varieties and its potential adulterants using a range of analytical techniques.

#### 3.3.2. Chromatographic

The chromatography technique is versatile and is used to detect adulteration in spices. Essential oils play a crucial role in the flavor and aroma of cinnamon. Chromatographic methods, such as GC, are used to quantify and analyze the essential oil composition, helping to distinguish between different cinnamon species.

The equipment consists of columns and mobile phases and can be used for separation in the gaseous state, gas chromatography (GC), or liquid chromatography (LC), with the aim of separating the constituents of a mixture of substances, either for identification, quantifying, or obtaining a pure substance [104].

By comparing the chromatographic profiles of authentic spices with potentially adulterated samples, chromatography can identify discrepancies in compound composition. It is possible to obtain precise quantification of specific compounds, helping to determine the extent of adulteration. In cases where pure compounds need to be isolated, chromatography can be used to separate and collect individual compounds from a mixture.

In numerous studies, high-performance liquid chromatography (HPLC) has been used to determine the cinnamaldehyde content in cinnamon samples. This compound is one of the primary components responsible for the characteristic flavor and aroma of cinnamon. Schad and Iwata [105] applied the HPLC method for the simultaneous analysis of two main compounds in ground cinnamon. A recovery test was performed by spiking coumarin and cinnamaldehyde in *C. cassia* and obtained results of 106 and 93.5%, respectively. In addition, a good repeatability was observed within the range of 0.013–1 mg/L for coumarin and 0.5–40 mg/L for cinnamaldehyde. Gursale et al. [106] use HPLC to quantitatively determine the levels of cinnamaldehyde and methyl eugenol within the methanolic extract derived from dried bark powder of *C. zeylanicum* Blume. It was observed that the mean amounts in bark powder were 8.76 mg/g cinnamaldehyde and 0.45 mg/g methyl eugenol.

Coumarin is a compound found in higher quantities in *C. cassia* compared to *C. verum*. HPLC, GC, and UPLC have been used to detect and quantify coumarin levels in cinnamon samples [107].

Balin and Sørensen [108] evaluated the amount of coumarin in food samples from traditional and/or seasonal bakeryware using UPLC. The authors verified that 18 samples showed values between 3.8 and 35.0 mg/kg, and they appointed the permissible amount of cinnamon that can be incorporated into food products without surpassing the EU regulations governing coumarin limits.

Chromatographic techniques, including GC and HPLC, are used to create chemical profiles, or fingerprints, of cinnamon samples. These profiles can be compared to authentic profiles to determine the genuineness of the cinnamon. Ananthakrishnan et al. [109] used the ultra-performance liquid chromatography-linear ion trap triple quadrupole mass spectrometry (UHPLC-ESI-QqQLIT-MS) method to determine the levels of coumarin and various phenolic compounds in genuine *C. verum* bark from South India. In authentic bark samples, the coumarin content ranged from 12.3 to 143.0 mg/kg, whereas market-sourced samples displayed considerably higher contents (>3462.0 mg/kg). The authors related that this significant increase in coumarin and cinnamaldehyde levels in market samples strongly indicated potential blending with substitutes such as *C. cassia* barks.

The UPLC-UV/MS was employed to assess the coumarin and cinnamaldehyde content in different samples, including authenticated cinnamon bark, locally purchased cinnamon, cinnamon-flavored foods, and cinnamon-based food supplements. The findings from the analysis revealed that *C. verum* bark contained minimal traces of coumarin, in contrast to the barks of *C. loureiroi* and *C. burmannii*, which contained significant and notable quantities of coumarin (0.31 to 6.97 g/kg). Cinnamaldehyde concentrations of 16.8 g/kg *C. verum*, 46.3 g/kg *C. burmannii*, 55.8 g/kg *C. loureiroi*, and 18.7 g/kg *C. cassia* were found [28].

Woehrlin et al. [110] evaluated cassia bark samples obtained directly from five trees of *C. cinnamon* from Indonesia using the HPLC method in relation to coumarin content. Remarkably, one of the trees displayed substantial variation in coumarin content, while no coumarin was detectable in the samples from two other trees. It was observed that there was significant variability in coumarin levels even within a single tree.

Pages-Rebull et al. [111] evaluated 87 samples, 16 of which were cinnamon, by HPLC-UV coupled with chemometrics (PCA, SIMCA, and PLS-DA), being compared in terms of sensitivity, specificity, and misclassification. Both methods provided similar results, with some sensitivity and specificity values below 1, and an overall error of 0.75% and 0.82% for PLS-DA and SIMCA. However, PLS-DA was more effective for characterization, identification, and authentication due to the lower overall error found. The chromatographic profiles obtained were significantly different from each other, making it possible to identify some biomarker compounds characteristic of cinnamon, such as eugenol and salicylaldehyde.

Ding et al. [112] used HPLC combined with PCA and PLS-DA on cinnamon bark and leaf samples collected from China, Vietnam, and Indonesia to assess properties and quality. The HPLC results showed good linearity, precision, and accuracy. Cinnamaldehyde was the most abundant marker component, boasting an average content of 86.25 mg/g. Trailing behind, eugenol exhibited a presence of 14.40 mg/g, while coumarin held its place with 5.79 mg/g. Cinnamyl alcohol contributed 1.13 mg/g to the composition, followed by cinnamic acid with 0.87 mg/g. These distinct quantities of compounds collectively form a signature chemical profile, revealing the intricate composition of the cinnamon samples under investigation. The similarity index and PLS-DA approaches discriminated cinnamon bark and leaf samples with a high accuracy of 98.2%.

Cuchet et al. [80] analyzed specific stable isotopes of the δ18 compound using gas chromatography-preparative isotope ratio mass spectrometry (GC-P-IRMS) to detect the addition of semi-synthetic compounds (carvone, (E)-cinnamaldehyde, and benzaldehyde) in cinnamon essential oils. The samples were collected from diverse botanical and geographical sources, such as *C. verum* from Sri Lanka and Madagascar, *C. cassia* from China, and *C. burmanii* Blume from Indonesia. No notable distinctions were detected between genuine and commercially available samples in relation to the carvone compound. The (E)-cinnamaldehyde found in cinnamon EOs, which were free from impurities, exhibited isotope ratios of δ13C at approximately −28.37 ± 1.14% and δ2H at around −142 ± 10%. The measured δ18O values exhibit significant variability, ranging from 1.2 to 21%. Isotope analyses conclusively confirm the presence of synthetic residue. These samples have experienced either partial or complete adulteration with (E)-cinnamaldehyde of synthetic or semi-synthetic origin.

Li et al. [44] analyzed the CEO composition of *C. verum*, C. *loureiroi*, and *C. cassia* species using GC-MS and FTIR. Trans-cinnamaldehyde was the major compound in the CEOs, covering a range from 66.28 to 81.97%. Among the analyzed species, *C. loureiroi* obtained the highest yield (3.08%). *C. verum* presented in its composition the eugenol compound in a marked way in relation to the other species, while *C. cassia* differed from the other species by presenting in its composition the α-guayene compound.

### 3.4. DNA Barcoding

It is a new biological identification method that makes use of DNA fragments as markers for species. In this method, a universal barcode is established, and the DNA data are analyzed and compared using a DNA code database to identify the analyzed species [113]. Some researchers used the chloroplast coding regions rbcL and *mat*K together with the *trn*H-psbA intergenic region as in DNA barcoding [114]. To evaluate intra- and interspecific genetic diversity of *Cinnamomum*, molecular identification can be used.

Abeysinghe et al. [115] identified many wild *Cinnamomum* species, such as *C*. *litseifolium, C*. *citriodorum*, *C*. *dubium*, *C*. *rivulorum*, *C*. *sinharajaense*, *C*. *ovalifolium*, and *C*. *verum* using the chloroplast regions, the *trnL* intron, the *trn*T*-trn*L, *trn*L*-trn*F, and *trn*H*-psb*A intergenic spacers, as the internal transcribed spacer (ITS) of nuclear ribosomal DNA (rDNA).

Doh et al. [116] evaluated *Cinnamomum* species used in traditional medicine. On the basis of the discrepancy in the determined ITS sequences, they developed a *C. cassia*-specific DNA marker using a 408-bp product amplified by the primer pair CC F1/CC R3. Using this DNA marker combined with the ITS-2 nucleotide sequence, products derived from Cinnamomum plants were monitored in markets in Korea, China, and Japan. In this work, most of the specimens monitored were derived from *C. cassia*.

Bhau et al. [117] standardized a protocol for extracting DNA from the genus *Cinnamomum* sp. Freeze-dried and fresh leaves of the *C. tamala* species were used to represent young and old leaves, in addition to *C. verum* and *C. zeylanicum* species. Twelve solutions for DNA protection with different concentrations were prepared. The protein solution composed of 0.7% activated charcoal, 2% CTAB, and 2% polyvinylpyrrolidone was the one that best extracted DNA. There was no difference in the quantity and quality of DNA extracted from freeze-dried or dried leaves. The standardization protocol was tested on four different plant species (*Androgrephis paniculata*, *Litsea cubeba*, *Azadirachta indica*, and *Cinnamomum camphora*). For all species, the protocol yielded high quality and quantity of extracted DNA being 501.66, 439.4, 341.36, and 317.4 ng/µL DNA, respectively. The protocol also worked for cinnamon species using *C. tamala*, *C. verum,* and *C. zeylanicum* of 246.82, 190.21, and 257.80 ng/µL, respectively. The primers used to verify the quality of *Cinnamomum* sp. were rbcL + matK and trnH-psbA. All primer blots in all *Cinnamomum* samples proved the high quality of the isolated DNA.

Tnah et al. [118] established a DNA barcode authentication and identification system for 112 plant species, of which two were cinnamon (*Cinnamon iners* and *C. verum*). DNA was extracted from leaves, amplified by PCR and sequenced for DNA barcoding markers using a layered multigene approach: rbcL for primary differentiation and trnH-psbA for secondary differentiation. Using rbcL, the generation of DNA barcodes was successful (100%), while for trnH-psbA, it had a rate of 96.4%. Barcoding gap and phylogenetic tree analyses demonstrated that rbcL provides apparent genus and species resolution and was able to separate most species.

Zhang et al. [119] used a DNA barcode identification method using the ITS2 and psbA-trnH sequences for 91 samples of powdered spices, of which five samples were cinnamon. The results using the DNA sequences indicated that only two spices were correctly labeled, and the other 14 spices showed different amounts of adulteration, making it possible to detect the presence of fennel in the cinnamon samples. The technique used proved viable when detecting adulteration in spices.

### 3.5. Electronic Nose

According to Gardner and Bartlett [120] an electronic nose is an instrument comprising a set of chemical sensors with partial specificity and an adequate pattern recognition system capable of considering simple or complex odors. The electronic nose is a system that mimics the structure of the human nose while trying to reduce its limitations.

In the human olfactory process, whenever the ortonasal pathway absorbs a volatile compound, it reaches the olfactory epithelium located in the upper nasal cavity, where interactions with receptor cells occur, and then different classes of olfactory neurons produce electrical stimuli that are transmitted to the brain. A pattern recognition process is generated by the memory so that it can then identify, classify, or perform a hedonic analysis. Evidence shows that a single olfactory neuron responds to multiple odorants, and each odorant is detected by multiple olfactory neurons. The electronic nose works using a series of sensors, this set of sensors has partial specificity and an appropriate system for pattern recognition. Usually, different sensors are used whose selectivity’s overlap for different molecules. In addition, the response of a sensor is commonly measured as a function of the change of some physical parameter, such as conductivity [121,122,123]. Figure 4 presents a comparison between the human olfactory system and the electronic nose.

Electronic noses can have different formats, but they all include the same basic elements, such as: I—aroma extraction system or sampling chamber consisting of a chamber in which the sample is packed before measurement; II—air flow unit generally composed of a synthetic air pump or cylinder responsible for generating air flow, subdivided into two independent outlets open alternately; III—sensitive chamber that contains the arrangement of gas sensors; IV—data processing and collection system that includes all electrical circuits and data acquisition software; and V—pattern recognition that consists of classifying volatile compounds according to the data stored during detection [122,124].

The electronic nose can be applied in the most diverse areas of industrial production and human activities, including medical diagnostics, environmental monitoring, and security systems [125]. Electronic noses offer precise, efficient, and non-invasive solutions to numerous challenges, making them indispensable in modern food science. Some key applications of electronic noses in the food industry can be related.

-Processing control and product uniformity. Maintaining consistent product quality is paramount in the food industry. Electronic noses enable real-time monitoring of aromas and odors during food processing, helping to ensure product uniformity. By providing instant feedback, they help manufacturers make timely adjustments to maintain desired product attributes.-Safety of working conditions. The safety of workers in food processing plants is of utmost importance. Electronic noses can detect harmful gases, such as ammonia or volatile organic compounds (VOCs), ensuring that working conditions remain safe. This technology helps prevent exposure to hazardous substances and enhances workplace safety.-Quality control assessments. Electronic noses are valuable tools for quality control assessments. They can identify subtle differences in aroma profiles, allowing for the precise determination of product freshness and overall quality. This is particularly important in industries where subtle variations greatly affect product value;-Contamination and maturation of food. Detecting contaminants or spoilage in food products is crucial to preventing health hazards and financial losses. Electronic noses can identify spoilage or contamination by recognizing changes in odor profiles. Additionally, they aid in monitoring the maturation process of foods such as fruits, helping to determine the optimal harvesting time;-Flavor/odor characteristics. Understanding and replicating desired flavors and aromas are essential in food manufacturing. Electronic noses assist in the analysis of complex flavor and odor characteristics, enabling the creation of products with consistent and appealing sensory attributes;-Geographical origin and confirmation of botanical origin. The origin of food products is a critical aspect of quality assurance and fraud prevention. Electronic noses can differentiate between products based on their geographical and botanical origins by analyzing unique odor fingerprints associated with specific regions or plant varieties;-Adulteration and authentication. Food fraud, such as the adulteration of high-value products, poses significant challenges to the industry. Electronic noses can quickly identify counterfeit or adulterated products by detecting inconsistencies in odor profiles, helping to ensure product authenticity;-Variety and aging. In industries such as tea, coffee, and spices, the variety and aging of products significantly impact their quality and value. Electronic noses play a vital role in characterizing these attributes, helping producers maintain product consistency and meet consumer expectations.

Different types of gas sensors with specificities (reaction and response to volatile compounds present in the samples) can be used in the electronic nose [126,127,128].

Gas sensors are devices chemically sensitive to volatile compounds capable of, based on interactions or chemical changes, operating based on changes in electrical values, such as electrical current, resistance, and impedance, among others [129]. They can be classified into different types depending on the materials used as the sensing layer. Table 4 presents a summary of the different gas sensors applied in the detection of volatile compounds.

Among these, the most commonly used are metal oxide semiconductor gas sensors that have good sensitivity and respond to oxidizing compounds, followed by conductive polymers that have low cost, good stability, and ease of synthesis [130,131,132,133].

Gas sensors are evaluated for their performance by different parameters such as sensitivity, selectivity, response time, return time, reversibility, manufacturing cost, stability [134], hysteresis [135], and detection limit. These parameters can be influenced according to the choice of construction methods and design of the electrodes, substrates, sensitive materials, synthesis methods, doping, and ways of deposition of the sensing layers, so an ideal sensor must have high sensitivity, selectivity, stability, a low response time and turnaround time, low sensitivity to temperature and humidity variations, and a low manufacturing cost [125,134].

From the fingerprint results combined with pattern recognition, it is possible to distinguish the volatile profiles and classify them according to the level or concentration of adulteration, type, technological process, and origin. Thus, the electronic nose plays an important role in quality detection, authentication, and food adulteration based on the analysis of volatile compounds, being widely used in different food matrices such as meats, oils, teas, coffees, grains, beers, milk, fruits, vegetables, and spices, among others, as can be seen in Table 5 [136,137,138].

The electronic nose is undoubtedly a promising and revolutionary tool in the detection and discrimination of adulterations in essential oils, foods, and drugs. Its ability to mimic the sensitivity of human smell, combined with the precision of chemical sensors and the speed of generating results, makes it a crucial technology in guaranteeing the quality and authenticity of these products.

Growing concern about adulteration in high-value products such as essential oils, as well as food safety and drug integrity, places the electronic nose in a prime position to meet these critical demands. Its sensitivity to identify adulterations even at very low levels, below 10%, is a significant advance in protecting consumers and preserving the integrity of the industry.

A commercial electronic nose composed of an array of 10 different oxide semiconductor sensors was used to identify and distinguish coumarin-enriched Japanese green tea. The array of gas sensors showed different responses to the particular (coumarin-like) flavor. Through PCA and cluster analysis, it was observed that there was a separation among the seven tea samples. Moreover, it highlights a distinct differentiation between the coumarin-enriched green teas and the group of green teas with lower coumarin content.

An electronic nose utilizing carbon nanocomposites has been developed for the detection of clove essential oil, eugenol, and eugenyl acetate. It is noteworthy that as the concentration of the volatile compounds increases, there is a proportional increase in the voltage response, indicating a linear relationship. This correlation is robust, with a correlation coefficient (R^2^) exceeding 0.99 across all sensors. Using PCA, a distinction was observed between the concentrations evaluated for both eugenol and eugenyl acetate [137].

Bio-active volatile terpenes and natural compounds such as *trans*-cinnamaldehyde were investigated by HS-SPME, followed by GC-MS, and with an electronic nose composed of an array of 18 gas sensors for the aroma pattern. Both techniques are able to detect the compounds; by PCA, it was possible to distinguish *trans*-cinnamaldehyde, where PC2 scores were gradually decreased by increasing terpene concentrations [153].

As electronic nose technology continues to evolve and become more accessible, we can expect its presence and application to expand further in key industries. Not only will this help prevent fraud and ensure product quality, but it will also have a positive impact on consumer confidence and public health.

## 4. Conclusions

Cinnamon, in the form of powder, bark, or CEOs, is used in several products. The varieties found on the market are classified as true cinnamon and false cinnamon. The true CEOs have a higher content of phenolic and aromatic compounds and a higher price compared to the false CEOs, making them more difficult to find in the market. As false CEOs present similar sensory characteristics to true ones, they can be used as adulterants. To detect compounds and adulterations in products, mainly cinnamon, there are many techniques, such as physical, sensorial, and analytical methods, with emphasis on the last one, which is more explored because it uses spectroscopic, chromatographic, DNA barcoding, and electronic nose techniques, either alone or in combination with the chemometrics technique. Each technique has its particularities, but it is fundamental that it should be robust, cost-saving, produce reliable and accurate results, and be environmentally friendly.

## Figures and Tables

**Figure 1 micromachines-14-01819-f001:**
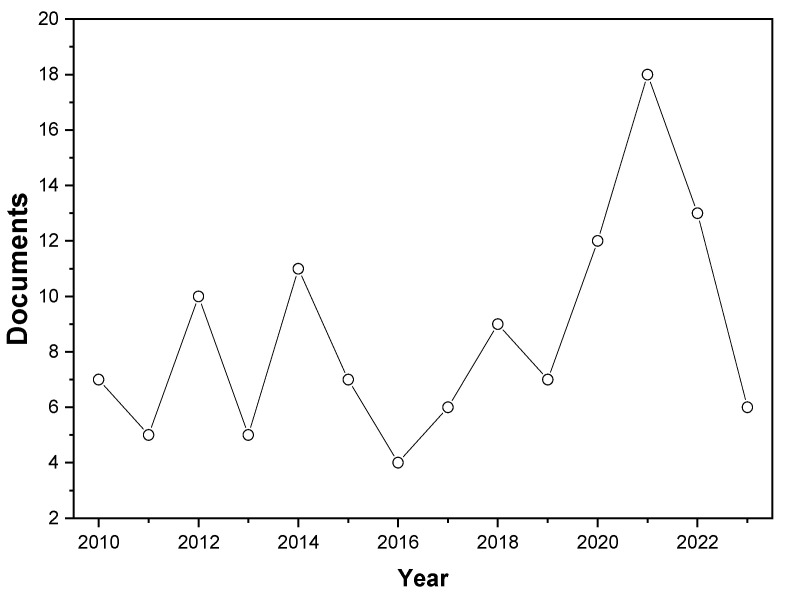
Number of annual peer-reviewed publications related to cinnamon.

**Figure 2 micromachines-14-01819-f002:**
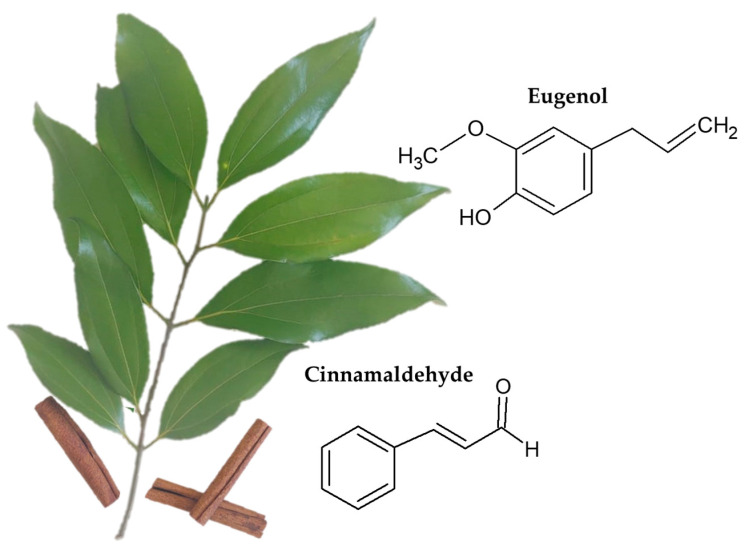
Parts of cinnamon and its main constituents.

**Figure 3 micromachines-14-01819-f003:**
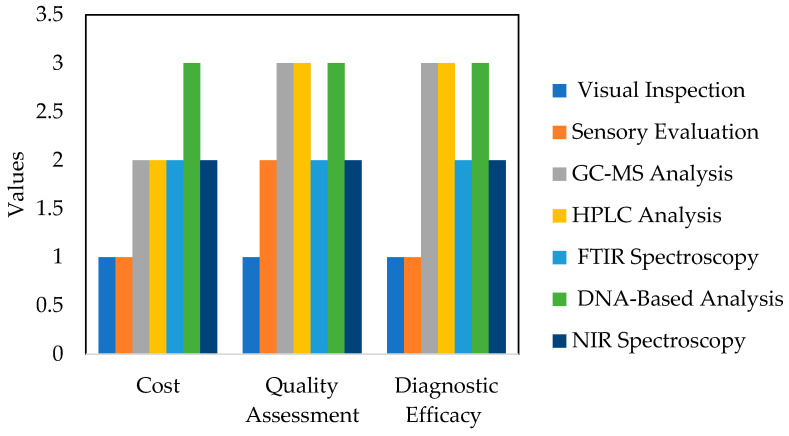
Comparison of detection methods for compounds and cinnamon adulterations.

**Figure 4 micromachines-14-01819-f004:**
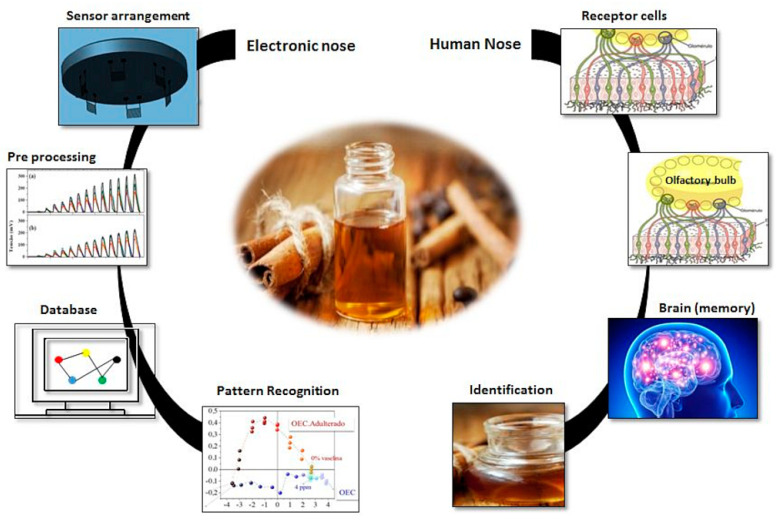
Similarity between the human olfactory system and the electronic nose.

**Table 3 micromachines-14-01819-t003:** Biological, insecticidal, and antidiabetic activity of cinnamomum.

Cinnamomum Species/Type	Sample	Biological Activity	Result	References
Insecticidal activity				
*C. verum*	Essential oil	*Odontotermes assamensis*	2.5 mg/g	[59]
*C. osmophloeum*	Essential oil	*Aedes albopictus*	40.8 µg/mL	[60]
		*Culex quinquefasciatus*	31.6 µg/mL	
		*A rmigeres subalbatus*	22.1 µg/mL	
*C. zeylanicum* L.	Essential oil	*Acanthoscelides obtectus*	46.8 µL/kg	[61]
*C. cassia*	Bark extract	*Tribolium castaneum*	3.96 µg/adult	[62]
		*Lasioderma serricorne*	23.89 µg/adult	
Antioxidant activity				
-	Cinnamon powder (raw extract)	ABTS	1.52 mg/mL	[63]
	Cinnamon powder (in vitro digestion)		1.18 mg/mL	
*C. cassia*	Extract	DPPH	10 mg/mL	[64]
*C. burmannii*	Essential oil (leave)	DPPH	100 µg/mL	[65]
*C. cassia*	Cinnamon bark oil	DPPH O_2_	10 mg/mL 1 mg/mL	[64]
*C. zeylanicum*	Essential oil (leave)	DPPH	4.78 μg/mL	[66]
		ABTS	5.21 μg/mL	
*C. zeylanicum*	Extract	ABTS	1119.9 µmol Trolox/g MS	[67]
		PCL	177.4 µmol Trolox/g MS	
		CV	39.8 µmol Trolox/g MS	
Anti-inflammatory activity				
*C. osmophloeum*	Essential oil (leave)	NO production in RAW 264.7 cells	9.7 to 65.8 µg/mL	[68]
*C. cassia*	Extract	NO production in RAW 264.7 cells	9.3 to 43 µg/mL	
Antidiabetic effect				
*-*	Gelatin capsule with cinnamon powder	Group A: placebo in capsule	17.4% reduction after 12 weeks	[69]
		Group B: 1000 mg/day of cinnamon powder in capsule form	10.12% reduction after 6 weeks	
*-*	Cinnamon extract	Rats divided into 5 groups (I—placebo only, II to V extract concentrations)	The highest dose of 200 mg/kg was more effective	[70]
*-*	Cinnamon Polyphenols	Mice divided into 5 groups (diabetic model, dimethylbiguanide, low, moderate and high dose of polyphenols)	Treatments with different doses of polyphenols (0.3, 0.6 and 1.2 g/kg/d caused a marked reduction in glucose	[71]
*C. osmophloeum*	Essential oil (leave)	Mice were induced with diabetes and then divided into six groups receiving different concentrations of essential oil	All doses tested significantly reduced blood glucose	[72]
Antimicrobial activity				
*C. verum*	Essential oil	*Staphylococcus hyicus*	minimal inhibitory concentration (MIC) and minimal bactericidal concentration (MBC) values ranging from 0.078 to 0.313%	[73]
*C. verum*	Essential oil	*Streptococcus suis* *Actinobacillus pleuropneumoniae*	MIC and MBC ranging from 0.01 to 0.156% (*v*/*v*)	[74]
*C. verum*	Essential oil	*Candida tropicalis*	MIC of 7.8 µL/mL	[75]
*C. cassia*	Essential oil	*Candida albicans*	65 μg/mL	[76]
*C. cassia*	Essential oil	*Staphylococcus aureus*	1.25% MIC	[77]
*C. cassia*	Essential oil	*Escherichia coli, Pseudomonas aeruginosa*, and *Streptococcus pyogenes*	0.25 to 0.50 mg/mL MIC	[78]
*C. cassia*	Essential oil	*Aspergillus flavus*, *Penicillium viridicatum*, and *Aspergillus carbonarius*.	1.67 to 5.0 µL/mL MIC	[79]

**Table 4 micromachines-14-01819-t004:** Types of gas sensors used in the electronic nose and their detection principle.

Types of Gas Sensors	Detection Principle
Acoustics	Frequency change
Calorimetric	Heat or temperature change
Catalytic	Electric field change
Colorimetric	Color change/absorption
Conductive polymers	Resistance change
Electrochemicals	Current or voltage change
Fluorescence	Fluorescent light emission
Infrared	Infrared radiation absorption
Metal oxide semiconductors	Resistance change
Optics	Light modulation, optical changes

Font: Wilson and Baietto [125].

**Table 5 micromachines-14-01819-t005:** Examples of gas sensors applied in the detection of adulteration, authenticity, and quality of food.

Analysis	Gas Sensor	References
Adulteration of lard with chicken fat	Quartz crystal	[139]
Adulteration of beef with pork	Colorimetric	[140]
Adulteration of beef with pork	Metal oxide semiconductor	[141]
Adulteration with inferior quality coffee	Metal oxide semiconductor	[142]
Adulteration of milk with formalin, hydrogen peroxide and sodium hypochlorite	Metal oxide semiconductor	[143]
Adulteration of ripe tomato juice with fresh tomato juice	Metal oxide semiconductor	[144]
Halal Authentication and Verification	Surface acoustic wave	[145]
Differentiation of Spanish wines	Quartz micro balance	[146]
Authenticity of geographic origin and classification levels of green tea	Colorimetric	[147]
Oxidized chicken fat flavor	Metal oxide semiconductor	[148]
Fresh meat quality	Metal oxide semiconductor	[149]
Diagnosis of apple juice contamination	Metal oxide semiconductor	[150]
Determine the concentrations of additives in tangerine juice	Metal oxide semiconductor	[151]
Evaluation of the quality of oil used for frying	Chemical sensor	[152]

## Data Availability

No new data were created.

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
