# Peer review of "Differentiating True and False Cinnamon: Exploring Multiple Approaches for Discrimination"

_micromachines, 2023, doi:10.3390/mi14101819_

Round 1

Reviewer 1 Report

Dear respected colleagues,

I have read your manuscript with great interest, and I found the topic and discussion to be intriguing. However, there are some comments that should be addressed to enhance the quality of this paper. Please check the following tips and amend the manuscript accordingly:

1.      Please check the reference list and make sure that problematic papers (e.g., retracted records) are not cited within the reference list.

2.      Please add DOI identifiers to all papers cited here.

3.      The manuscript text requires English editing.

4.      Line 17: The authors mentioned that they conducted a comprehensive literature review to discuss the current topic. However, it is not clear whether the respected authors followed any standard literature review procedure such as PRISMA or other methodologies that were used to conduct such studies. Please clearly mention which protocol was used for the literature review, which keywords were used to search the literature, how many papers were removed from searches, which scientific databases were searched, and how the results were interpreted. I highly recommend you add these cases to the M&M section. Please also try to use VOSviewer, a literature search software, to know in which fields the majority of studies have been conducted on your topic. You can use the output of VOSviewer to manage your discussion and enhance the quality of your paper.

5.      For all figures that have been taken from external sources such as websites, blogs, tweets, papers, and so on, the respected authors should provide a permission letter for the journal. Please use the available figures in the literature carefully.

6.      Please merge figures 1-4 into one figure. Since these figures were repeatedly reported in the literature, it is not necessary to consider such figures in this intriguing review paper. The respected authors can use Figma or PowerPoint to draw vector-based and high-quality figures for this study.

7.      Please use the same size fonts for labels represented in Figure 5. Please also add some more details to its caption.

8.      Please summarize some sections of the manuscript into a table, especially when you discuss the biological activity of the discussed plant. Any biological activity discussed here should be linked to the numerical values recorded for that activity. For example, if secondary metabolites of the discussed plant showed anti-insect activity, the respected authors should add the percent of inhibition, IC50 values, and so on to determine which compounds of the discussed plants have a higher inhibitory or biological activity profile.

9.      The respected authors can add a flowchart to section 2.2 where they discussed the adulteration in cinnamon. You can classify the detection methods based on their cost, quality, and diagnostic efficacy to provide excellent information for academic authors who want to read your paper and select a method for experimental assays.

10.   Please clearly mention which type of adulteration may occur in the discussed plant products and how regulatory sectors can prevent the entrance of adulterated products into local markets.

Rasouli. H

English requires further edition

Author Response

Response to the Reviewers comments

We are grateful to the editor and reviewers for their time and excellent comments on our manuscript. The recommended changes have been made to the manuscript and have been highlighted as requested.

Below, we provide a point-by-point response explaining each of the reviewer’s comments.

Response to Reviewer 1:

I have read your manuscript with great interest, and I found the topic and discussion to be intriguing. However, there are some comments that should be addressed to enhance the quality of this paper. Please check the following tips and amend the manuscript accordingly:

  1. Please check the reference list and make sure that problematic papers (e.g., retracted records) are not cited within the reference list.

Answer: The refenrences were revised.

  1. Please add DOI identifiers to all papers cited here.

Answer: All the references was checked, and when DOI was available this was included in the reference.

  1. The manuscript text requires English editing.

Answer: The English was revised.

  1. Line 17: The authors mentioned that they conducted a comprehensive literature review to discuss the current topic. However, it is not clear whether the respected authors followed any standard literature review procedure such as PRISMA or other methodologies that were used to conduct such studies. Please clearly mention which protocol was used for the literature review, which keywords were used to search the literature, how many papers were removed from searches, which scientific databases were searched, and how the results were interpreted. I highly recommend you add these cases to the M&M section. Please also try to use VOSviewer, a literature search software, to know in which fields the majority of studies have been conducted on your topic. You can use the output of VOSviewer to manage your discussion and enhance the quality of your paper.

Answer: The authors thanks for this suggestion. We included in the revised manuscript this information about 1 upward trajectory in scientific publications related to cinnamon over time.

  1. For all figures that have been taken from external sources such as websites, blogs, tweets, papers, and so on, the respected authors should provide a permission letter for the journal. Please use the available figures in the literature carefully.

Answer: The authors revised all Figures and only used them with permission.

  1. Please merge figures 1-4 into one figure. Since these figures were repeatedly reported in the literature, it is not necessary to consider such figures in this intriguing review paper. The respected authors can use Figma or PowerPoint to draw vector-based and high-quality figures for this study.

Answer:  The Figures were changed or removed.

  1. Please use the same size fonts for labels represented in Figure 5. Please also add some more details to its caption.

Answer:  The Font was standerized and provided more informations.

  1. Please summarize some sections of the manuscript into a table, especially when you discuss the biological activity of the discussed plant. Any biological activity discussed here should be linked to the numerical values recorded for that activity. For example, if secondary metabolites of the discussed plant showed anti-insect activity, the respected authors should add the percent of inhibition, IC50 values, and so on to determine which compounds of the discussed plants have a higher inhibitory or biological activity profile.

Answer: The authors provide a Table summaring the biological activity of cinnamomum plant.

  1. The respected authors can add a flowchart to section 2.2 where they discussed the adulteration in cinnamon. You can classify the detection methods based on their cost, quality, and diagnostic efficacy to provide excellent information for academic authors who want to read your paper and select a method for experimental assays.

Answer: The authos provide a Figure with a summarization of the methods applied to detect compounds and cinnamon adulterations based on cost, quality, and diagnostic efficacy.

  1. Please clearly mention which type of adulteration may occur in the discussed plant products and how regulatory sectors can prevent the entrance of adulterated products into local markets.

Answer: The authors included more informations about the types of adulterations and entrance of adulterated.

Reviewer 2 Report

The authors provided a comprehensive literature review investigating the distinctions between true and false cinnamon.This review is well written as a whole. However, some small details need to be revised. Therefore, I recommend it to be accepted after a minor revision.

The specific content is as follows:

1. The author's unit recommends revising according to the requirements of the journal.

2. Essential oils (EOs) was the first full name (abbreviation), and then uniformly abbreviated the word. Such as Line 43 and 62.

3. 2. Cinnamomum sp. Should it be oblique format?

4. The quoted picture authorization format is strictly revised in accordance with the requirements of the journal, and authorization is required. For example, Adapted with permission from Redação GreenMe Canela [19]. Copyright 2022,Publisher name. Figure 1, 2, 3,4 

Author Response

Response to reviwer 2:

 The authors provided a comprehensive literature review investigating the distinctions between true and false cinnamon.This review is well written as a whole. However, some small details need to be revised. Therefore, I recommend it to be accepted after a minor revision.

The specific content is as follows:

  1. The author's unit recommends revising according to the requirements of the journal.

Answer: The manuscript was revised.

  1. Essential oils (EOs) was the first full name (abbreviation), and then uniformly abbreviated the word. Such as Line 43 and 62.

Answer: Thanks for the suggestion. It was revised.

  1. 2. Cinnamomum sp. Should it be oblique format?

Answer: It was revised

  1. The quoted picture authorization format is strictly revised in accordance with the requirements of the journal, and authorization is required. For example, Adapted with permission from Redação GreenMe Canela [19]. Copyright 2022,Publisher name. Figure 1, 2, 3,4.

Answer: The authors revised the permission of all Figures.

Round 2

Reviewer 1 Report

I have no further comments on this paper. I highly recommend the respected authors to re-revise the manuscript before publication because it has 29% similarity with previously published papers. Please check the attached file and revise the manuscript accordingly. 

Best regards, 
